# Invasive Aspergillosis in the Intensive Care Unit

**DOI:** 10.3390/jof11010070

**Published:** 2025-01-17

**Authors:** Anna Zubovskaia, Jose A. Vazquez

**Affiliations:** Division of Infectious Diseases, Medical College of Georgia, Augusta University, Augusta, GA 30912, USA; azubovskaia@augusta.edu

**Keywords:** aspergillosis, intensive care unit, invasive aspergillosis, antifungals

## Abstract

Invasive aspergillosis (IA) is a fungal infection, which has traditionally been associated with neutropenia and immunosuppressive therapies. Our understanding of invasive aspergillosis has been evolving and, in the past few decades, IA among ICU patients has been recognized as a common infection and has become more widely recognized. The diagnosis and management of invasive aspergillosis in the ICU is particularly challenging, due to the unstable clinical condition of the patients, lack of diagnostic markers, increased risk of further clinical deterioration, multiple comorbidities, and a need for early assessment and treatment. In this article, we will discuss the challenges and pitfalls of the diagnosis and management of invasive aspergillosis in an ICU setting, along with a review of the current literature that is pertinent and specific to this population.

## 1. Introduction

Invasive aspergillosis (IA) is an infection caused by a saprophytic filamentous fungus of the *Aspergillus* species [1]. It is a known opportunistic infection in immunocompromised hosts, such as patients with acute myelogenous leukemia (AML), prolonged neutropenia, solid organ transplant (SOT) recipients and allo-stem cell transplant recipients (allo-SCT) [2]. However, invasive aspergillosis is an increasingly recognized infection among patients in the intensive care unit, often developing in patients without classic risk factors and complicating other respiratory diseases such as COPD, severe COVID-19 and influenza [1,2,3]. Patients with COVID-19-associated pulmonary aspergillosis (CAPA) and influenza-associated pulmonary aspergillosis (IAPA) have a higher mortality than patients without CAPA/IAPA [3,4,5]. In fact, when compared to other respiratory tract infections, the mortality among patients in the ICU who developed IA can be as high as 46% [6]. The diagnosis of invasive aspergillosis and critically ill patients remains challenging. In this review, we will discuss the epidemiology of IA among ICU patients and the current evidence-based approach to the diagnosis and management of IA in the ICU, including IAPA and CAPA.

## 2. Epidemiology

The diagnostic challenges and the low suspicion among providers make IA a difficult diagnosis to establish and thus, make the true incidence of IA among ICU patients difficult. However, the prevalence of IA among severely ill patients and ICU patients seems to be increasing including patients with COVID-19, influenza virus and COPD. Solid organ transplant status, solid organ tumors and prolonged exposure to corticosteroids, as well as the use of ibrutinib, are increasingly recognized as risk factors for invasive aspergillosis.

The incidence of invasive aspergillosis in the intensive care unit has been reported to be between 0.33% and 7% [1,2,5,6,7,8,9]. Reviewing the service level database including inpatient data from the years 2005–2008, 6.4% of ICU patients were diagnosed with IA [6]. In this study, evaluating the diagnostic accuracy of the Asp-ICU algorithm, 71% of patients with proven invasive aspergillosis were immunocompromised and 70% of patients with putative IAPA had known risk factors [10]. *Aspergillus fumigatus* is the most commonly encountered species among the ICU patients, isolated in 92% of the cases, followed by *A. flavus*, *A. niger* and other species [11]. *A. fumigatus* was also the most frequently encountered species among patients with IAPA, as well as CAPA [12,13]. In this study, the predictive score for the development of CAPA was evaluated and *A. fumigatus* was isolated in 64.3% of cases, followed by *A. niger* complex, less frequently *A. terreus* and *A. flavus* [13].

Influenza-associated pulmonary aspergillosis (IAPA) is a complication of severe influenza which causes increased morbidity and mortality compared to patients without IAPA. The incidence of IA ranges between 10 and 32% of influenza patients admitted to the ICU [14,15,16].

Invasive aspergillosis-complicating COVID-19 infection (CAPA) is an increasingly recognized phenomenon. Based on the published data, the prevalence of CAPA has been estimated to be between 5 and 30% [17].

Increased mortality among patients with invasive aspergillosis in the ICU, IAPA and CAPA has been described in multiple studies [2,16,17,18].

## 3. Risk Factors and Pitfalls in IA Definitions in the ICU

Known and classic risk factors of IA include immunosuppression, such as prolonged neutropenia due to leukemia or other hematological malignancy, because of chemotherapeutic agents or as a result of HSCT, as well as acute GVHD grade 3 or 4, involving either the gastrointestinal tract, lungs or liver that is refractory to first-line treatments such as steroids. The duration and severity of neutropenia are also associated with an increased risk of IPA [19,20]. Solid organ transplant recipients (specifically, lung and heart transplant recipients) and the prolonged use of corticosteroids (equal to or above 0.3 mg/kg corticosteroids for equal to or above 3 weeks in the past 60 days) are other well-known predisposing conditions [1,21]. Additional risk factors for the development of IA include treatment with T-cell immunosuppressants, such as calcineurin inhibitors, TNF alpha blockers, lymphocyte-specific monoclonal antibodies, or immunosuppressive nucleoside analogs for 90 days. In addition, treatment with a recognized B-cell immunosuppressant, such as Bruton’s tyrosine kinase inhibitors (ibrutinib), has also been associated with IA. Inherited severe immunodeficiencies, such as chronic granulomatous disease, STAT3 deficiency or SCID (severe combined immunodeficiency), can become predisposed to invasive pulmonary mold disease.

These are uniformly recognized host risk factors included in the consensus definition of invasive fungal disease published by EORTC/MSG [21]. However, patients in the ICU setting lack traditional risk factors and different pathogenic mechanisms predispose them to invasive aspergillosis in the ICU setting (Table 1) [11]. This poses a significant diagnostic challenge since the ICU patients who do not meet the EORTC/MSG criteria cannot fulfill the criteria used to diagnose invasive aspergillosis. Furthermore, tissue sampling in these settings may be difficult or contraindicated in patients in the ICU with hemodynamic instability, coagulopathy or thrombocytopenia. Moreover, the classic radiographic signs of IA, such as the halo sign and air crescent sign, are infrequently encountered in patients without classic risk factors (most ICU patients) and *Aspergillus* serum galactomannan testing in this population has been shown to have low sensitivity and specificity. In addition, the low importance of invasive aspergillosis in the ICU setting represents another diagnostic challenge [22].

The Asp-ICU score was developed to capture cases of IAPA in non-neutropenic ICU patients who could not fulfill the criteria classic EORTC/MSG criteria [10]. Autopsy results among deceased ICU patients indicated that adherence to host factors for invasive fungal disease as one of the key features used to meet the criteria for invasive fungal disease increased the risk of a missed diagnosis [8]. High mortality rates among patients with CAPA/IAPA in the ICU have been found in the absence of traditional risk factors, such as neutropenia, SCT or other types of immune deficiency. The ASP ICU diagnostic algorithm was created based on EORTC/MSG criteria for invasive fungal disease; however, the interpretation of imaging and microbiological data was modified to increase its utility in diagnosing IA specifically among ICU patients [8]. The entry criterion was positive lower respiratory culture, and the probability of a true invasive infection was assessed by evaluating the clinical and radiological data in conjunction with the positive culture. Clinical criteria, such as recrudescent fever despite at least three days of appropriate antimicrobial therapy and no other apparent source, as well as worsening respiratory insufficiency despite appropriate antibiotic therapy and ventilatory support, were specifically tailored to ICU patients. The radiological criterion (abnormal X-ray or CT findings) appreciated the frequently nonspecific imaging findings among ICU patients with IA and low sensitivity of the classic radiological signs, such as the air crescent sign or halo sign in the ICU population, and increased the sensitivity of the algorithm among the ICU cohort. Finally, the presence of the classic host factors, traditionally considered a prerequisite for the diagnosis of IA, although still present among the criteria, was no longer a requirement [8]. The algorithm demonstrated a higher sensitivity and negative predictive value in patients without classic risk factors as well as in immunocompromised hosts [10].

The original ASP ICU algorithm proposed over a decade ago did not include fungal biomarkers, such as galactomannan and the *Aspergillus* quantitative PCR. A revision of the algorithm was later proposed to improve the detection of probable invasive pulmonary aspergillosis in the ICU (Table 2) [23]. These revisions are now included in the mycological criteria and the definitions of invasive pulmonary aspergillosis in ICU patients, including IAPA, CAPA and invasive fungal disease from EORTC/MSG (Table 2) [21,23].

The current diagnostic classification of invasive fungal disease according to the scale of certainty still requires the positive culture of the *Aspergillus* species along with a histopathological examination of the tissue obtained in a sterile fashion from a normally sterile site and with evidence of accompanying tissue damage. However, as previously discussed, such invasive methods are not always appropriate in the ICU setting. Thus, the diagnosis of probable invasive aspergillosis appears to be a more applicable criterion in this type of setting (Table 3).

Considering the challenges in diagnosing and defining cases of IA in an ICU setting, as well as the varying prevalence of IA across hospitals and the varying frequency of proven diagnoses, revised definitions of proven and probable invasive pulmonary aspergillosis in the ICU setting were proposed [22]. Revised definitions incorporated mycological evidence, clinical/radiological abnormality (see Table 2, sections regarding pulmonary aspergillosis and tracheobronchitis) and host factors. Mycological evidence for proven and probable IA was proposed. Additional host factors have been proposed to establish the diagnosis of IA in the ICU population. These risk factors reflect the unique pathogenesis of IA in the ICU population, in addition to the classic risk factors, such as SOT, neutropenia, hematological malignancies and HSCT. These additional risk factors include chronic respiratory airway abnormalities such as COPD or bronchiectasis; decompensated cirrhosis; HIV; treatment with recognized immunosuppressants (e.g., calcineurin or mammalian target of rapamycin [mTOR] inhibitors, blockers of tumor necrosis factor [TNF] and similar antifungal immunity pathways, alemtuzumab, ibrutinib, or nucleoside analogs) during the previous 90 days; glucocorticoid treatment with prednisone equivalent of 20 mg or more per day; and severe viral pneumonias such as influenza and COVID-19 [22]. Additionally, patients with severe burns, a prolonged ICU stay, and a positive fungal culture are also at risk for developing IA in the ICU [24].

Invasive aspergillosis-complicating viral pneumonia, such as influenza (IAPA) or COVID-19 (CAPA) is not a rare infection encountered among ICU patients. According to the study by Waldeck et al., 10.8% of 158 patients hospitalized in tertiary hospitals in Switzerland during the 2017/2018 and 2019/2020 influenza seasons who were admitted with a PCR were confirmed to have an influenza infection, required ICU admission for over 24 h and developed IAPA. In addition, those with a prior history of asthma and days of mechanical ventilation were also associated with the development of IAPA [16]. Several other studies show varying rates of IAPA up to 28.1%. Influenza was found to be an independent risk factor in the development of IA, conferring a high mortality among ICU patients [12,15,25,26].

*Aspergillus* hyphae are present in the airways and frequently represent colonization [20]. The key feature that determines the pathogenesis of invasive aspergillosis is the transition from colonization to invasion of the underlying tissues, specifically angioinvasion [27]. In the susceptible host, such as patients in the ICU or those with a viral infection of the upper respiratory tract, increased production of inflammatory cytokines leads to local tissue damage, which aids in further tissue invasion by *Aspergillus*. Several research groups suggested that respiratory viruses, such as SARS-CoV-2 and influenza, cause lung hyperinflammation, which causes defects in several levels of antifungal immunity, including the phagocytosis of *Aspergillus* conidia, the epithelial barrier and function and the neutrophil-mediated killing of *Aspergillus* hyphae [28]. In COVID-19, a decrease in T-cell populations, especially in patients with severe disease, has been observed, probably accompanied by defective lymphocyte function, increasing the risk of invasive mold infection [29,30]. Angioinvasion occurs in the pulmonary vasculature, as well as in other organs during the disseminated infection, and results in thrombosis and tissue infarction, as well as reduced leucocyte entry into the infected area and reduced delivery of antifungals [27].

There is a significant variation in estimates of the incidence of COVID-19-associated pulmonary aspergillosis (CAPA). According to the study by Hurt et al. conducted across five UK hospital intensive care units among patients admitted for mechanical ventilation or ECMO due to respiratory failure from COVID-19, the incidence of probable CAPA was 10.9%, followed by 5.2% of the study population with possible CAPA. No definitive cases of CAPA were found in this study [31]. In a multinational observational study conducted by the European Confederation of Medical Mycology (ECMM), the median prevalence of CAPA per enrolled center was 10.7% (range 1.7–26.8%) [18]. However, in the review of the autopsy series, which included case studies describing autopsies from 677 subjects, of which 320 were on mechanical ventilation, only 11/677 (2%) had an invasive mold infection. This included eight cases of CAPA, an unspecified number of invasive mold diseases and one disseminated mucormycosis. Among those who received mechanical ventilation (320/677), only 6 (2%) had an invasive mold infection [32].

In a retrospective cohort study from Johns Hopkins, the author’s goals were to design a CAPA prediction model using mechanically ventilated patients to stratify which patients were at risk of developing CAPA and who would benefit from additional testing and antifungal treatment. Age, time from intubation, use of dexamethasone for COVID-19 treatment, underlying pulmonary circulatory disease, multiple myeloma, cancer or hematologic malignancy were identified as risk factors for CAPA and were included in the prediction model. In that study, 11.8% (98 patients) of the cohort met criteria for CAPA. Patients with CAPA in this cohort had a higher mortality or were more likely to require advanced life support and to have a longer duration of advanced life support therapy [17].

Another prediction model was developed based on the retrospective matched case–control study conducted at a tertiary care center in South India [33]. The European Organization for Research and Treatment of Cancer Risk Factors used this prediction score to identify lymphopenia and broad-spectrum antimicrobials as the main risk factors for CAPA. Of note, due to low enrollment numbers, the study failed to identify hematologic malignancies, SOT and T-cell and B-cell immunosuppressants, as well as the use of ibrutinib, as risk factors for the development of CAPA. The recovery of bacterial pathogens from blood and respiratory secretions frequently decreased the risk of CAPA. All cases in the study were either probable or possible CAPA. In addition, the study included both mechanically ventilated and nonventilated patients. The diagnosis of CAPA increased the mortality rate and ICU admission; however, it did not significantly increase the risk of requiring mechanical ventilation. The isolation of bacterial pathogens in blood or BAL decreased the likelihood of CAPA. The sensitivity of this model was 77.4% with a specificity of 78.1%. Despite certain limitations, this model might hold promise in low-resource settings [33].

Another study published in 2024 by Iacovelli et al. evaluated risk factors for the development of CAPA in a respiratory sub-intensive care unit and its impact on overall mortality. Hematological malignancy, lymphocytopenia and COPD were identified as independent risk factors for CAPA. Furthermore, being over 65 years was identified as a predictor of mortality [34].

Another CAPA clinical prediction score based on a study by Calderón-Parra et al. from Spain, identified old age, active smoking, chronic respiratory disease, chronic renal disease, chronic corticosteroid treatment, tocilizumab therapy and a high Apache 2 score on admission as risk factors for CAPA [13].

In the recently published meta-analysis by Gioia et al., nine risk factors for CAPA were identified including chronic liver disease, neurological malignancy, chronic obstructive pulmonary disease, cerebrovascular disease and diabetes, as well as mechanical ventilation, the use of renal replacement therapy, the treatment of COVID-19 with interleukin-6 inhibitors and the treatment of COVID-19 with corticosteroids. Patients with CAPA were typically older and the duration of mechanical ventilation was longer among patients with CAPA than those without CAPA [35].

## 4. Clinical Presentation

The clinical presentation of invasive aspergillosis in the ICU setting frequently overlaps underlying co-existing comorbidities, such as COVID-19, influenza, pneumonia, sepsis, etc., which poses an additional diagnostic challenge [5,20]. Fever, cough, hemoptysis and a pleuritic chest can be the presenting symptoms of invasive pulmonary aspergillosis; however, these symptoms are nonspecific and might not be present in all cases [1,7,20]. The development of sinonasal or CNS aspergillosis might also be accompanied by the signs and symptoms of local or regional spread of the infection. There are significant differences in the clinical presentation between neutropenic and non-neutropenic patients [36]. Specifically, IA in non-neutropenic patients is generally less symptomatic and frequently associated with pneumonia due to another organism (viral) and associated with higher mortality rates [36]. Moreover, IA in non-neutropenic patients who had high serum GM levels were more likely to have COPD and had more severe respiratory symptoms, which included hemoptysis and dyspnea [37].

## 5. Diagnosis of Invasive Aspergillosis in the ICU

Histopathological diagnosis is based on the presence of tissue invasion and *Aspergillus* growth from the normal sterile site (Figure 1a,b) [38].

However, despite being the gold standard, positive *Aspergillus* cultures used as a diagnostic method have a very low diagnostic yield. This is even lower among non-neutropenic ICU patients who lack traditional host risk factors. In addition, microscopy and culture alone cannot distinguish between colonization and infection [39,40]. As previously discussed, biopsies obtained from sterile sites are required to fulfill the histopathological criteria of proven invasive aspergillosis. These procedures may be not feasible among some ICU patients due to unstable clinical conditions and contraindications to invasive procedures [2,22]. Due to these limitations, a less invasive approach to establishing a diagnosis is usually utilized [22]. Frequently, a diagnosis is presumptive and made based on imaging findings, serum biomarkers, sputum and/or BAL specimens and utilizes nonculture-based methods [22]. The sensitivity of BAL cultures differs between neutropenic and non-neutropenic patients [36].

Galactomannan is an *Aspergillus*-specific antigen, a major component of the *Aspergillus* cell wall. It can be measured in serum, plasma, BAL samples or CSF. Studies investigating the utility of galactomannan testing on upper airway samples have been previously published [38,39]. During the COVID-19 pandemic, alternative samples from the upper respiratory tract, such as non-bronchoalveolar lavage, tracheal aspirate and sputum, were suggested as an alternative to the more invasive BAL samples for obtaining Aspergillus cultures and galactomannan due to the restrictions imposed by the COVID-19 pandemic. However, these biomarkers have not been validated in tracheal aspirates, sputum and non-bronchoalveolar lavages. Since cut-off values have not been well established, results have to be interpreted with caution [41,42,43]. The angioinvasion of *Aspergillus* can result in galactomannan being released into the bloodstream and becomes detectable in serum or plasma. However, it is often not found in the serum of non-neutropenic patients where more airway invasions rather than angioinvasion are present [43,44]. The performance of the serum GM is also negatively affected by concurrent systemic anti-mold therapy, which may yield false-negative results [45]. False-positive results of the serum galactomannan assay have been described in patients who received certain antimicrobials, such as amoxicillin/clavulanate, piperacillin tazobactam and cefepime. False-positive results of galactomannan in the BAL have also been reported in patients receiving carbapenems and ceftriaxone [46]. Among patients post-HSCT within the last 100 days and in those with GI GVHD, false-positive results can be encountered due to compromise of the mucosal barrier and the translocation of the galactomannan through the intestinal mucosa [47]. Other fungal species (*Penicillium* spp., *H. encapsulatum*, and *Geotrichum* spp., etc.), containing galactomannan in their cell walls can also yield false positive results [48]. Among non-neutropenic patients, a positive GM in serum is encountered less often and has lower sensitivity, most likely due to less prominent angioinvasive features in these cases [29,37].

*Aspergillus* PCR in blood, serum or BAL were included in the mycological criteria and may be a useful tool to establish the diagnosis of invasive aspergillosis. According to the 2019 meta-analysis published by Cruciani et al., pooled sensitivity and specificity of PCR from blood is reported to be between 79% and 80% for a single positive result and 60% and 94% for two consecutive positive test results in immunocompromised people [49]. The performance of PCR in blood decreases in the setting of systemic mold-active therapy, as well as in non-neutropenic patients [50,51].

*Aspergillus* PCR in BAL is a promising tool for diagnosing IA in neutropenic and non-neutropenic patients, and it is generally not affected by antifungal therapy [52]. Various assays demonstrate varying degrees of sensitivity in immunocompromised patients [53,54]. However, it appears to have excellent specificity in critically ill patients with and without COVID-19 and/or immunocompromising conditions; although, the study by Mikulska et al. showed that sensitivity is higher among immunocompromised patients [54]. The sensitivity and specificity improve when several diagnostic tests are used in combination, including BAL-GM, BAL-PCR, serum GM and BAL cultures [55].

In recent years, the utility of metagenomic next-generation sequencing (NGS) and cell-free DNA is being actively explored, both in patients with hematological disorders and in those without immunocompromising conditions [56]. Next-generation sequencing of microbial cell-free DNA using the Karius test on plasma as compared to the standard of care procedures revealed a sensitivity of 38.5% and a specificity of about 97% in patients with probable IA. In addition, the results are affected by antimould therapy [57]. *Aspergillus* plasma cell-free DNA PCR showed superiority over *Aspergillus* serum galactomannan in the diagnosis of invasive aspergillosis among patients with hematological illnesses/stem cell transplants, demonstrating an overall sensitivity of 86% and a specificity of 93.1% (Table 4) [58].

The utility of metagenomic next-generation sequencing for the diagnosis in non-neutropenic patients at risk of invasive aspergillosis, including CAPA, is being explored [59,60,61,62,63]. The utility of blood biomarkers such as galactomannan is often poor in CAPA, as well as invasive pulmonary aspergillosis in non-neutropenic patients, as it exhibits early tissue-invasive growth in the lungs with delayed angioinvasion. The Karius test showed promising performance, with a high specificity of 97% [60].

More recently, the lateral flow assay (LFA) and the lateral flow device (LFD) have become point-of-care diagnostic tests for the diagnosis of invasive aspergillosis. They show good performances on both serum samples and BALF and aid in prompt diagnosis of IA. GM-LFA shows excellent performance in patients with hematological malignancies [64]. The performance in SOT recipients can be variable [65]. Several LFDs are being studied for clinical use among neutropenic patients, as well as SOT recipients and patients in the ICU [66,67,68].

**Table 4 jof-11-00070-t004:** Utility of mcfDNA (microbial cell-free DNA) analysis and NGS in diagnosis of invasive pulmonary aspergillosis.

Study, First Author	# of Patients	Population	Tested Sample	Sensitivity	Specificity	PPV	NPV	Additional Data
Liu, 2024 [61]	N = 66, 21 with IPA, 45 non-pulmonary aspergillosis	Patients with T2DM, with and without immunocompromising conditions	BALF (90.5%), blood (9.5%)	66.7%	100%	100%	86.5%	Significantly improved performance when combined with other methods
Bao, 2022 [62]	N = 33, 12 with IPA, 21 nonpulmonary aspergillosis	Non-neutropenic patients	BALF: N = 27 Blood: N = 6 Pleural fluid N = 1 BALF+serum: N = 3	91.7%	71.4%	64.7%	93.8%	Additionally evaluated co-pathogens in mixed infections, performance reported as cumulative for mNGS on all types of samples
Huygens, 2024 [57]	N = 106, proven/probable IA: N = 35, IA+other IFD: N = 4, other IFD = 7, possible IFD = 48	AML, MDS, HSCT, hematological malignancies, neutropenia	Karius test—plasma: N = 106, research-only pipeline Karius on BALF, N = 34	Plasma: 44% RUO-BAL KT: 72.2%	Plasma:96.6% RUO-BAL KT: 88.2	NA	NA	Data for multiple IFD, performance not impacted by mold active therapy
Lee, 2024 [63]	N = 34, N = 1 (proven IA), N = 25 (probable), N = 3 (putative), N = 5 (no IA)	Hematological malignancies = 16, COVID-19 = 19.	Plasma cfDNA, N = 34	N/A	N/A	N/A	N/A	Concordance between cfDNA and conventional methods of diagnosis is higher in HM group than in COVID-19
Hoenigl, 2023 [60]	N = 114, CAPA: proven: N = 0, probable: N = 6, possible: 2, No CAPA: N = 106	COVID-19 associated ARF in the ICU	Karius, plasma mcfDNA	67% for Aspergillus, 83% for other molds	97% for patients without CAPA	N/A	N/A	
Mah, 2023 [58]	N = 238, Proven: N = 15, probable: N = 31, possible: N = 62, no IA: N = 130	89.9% immunosuppressed (HM, HSCT, SOT)	Plasma cfDNA	Overall, 80.0% (varies in different groups)	Overall, 93.1%	5% prevalence: 39.6% 20% prevalence: 75.7%	5% prevalence: 99.2% 20% prevalence: 96.4%	Highest performance in HM/HSCT patients, performance characteristics vary by patient groups

Abbreviations: AML—acute myelogenous leukemia; ARF—acute respiratory failure; BALF—bronchoalveolar lavage fluid; CAPA—COVID-19-associated pulmonary aspergillosis; HM—hematological malignancy microbial cell-free DNA; mNGS—metagenomic next-generation sequencing; RUO-BAL KT—research-use-only Karius test on bronchoalveolar lavage fluid; SOT—solid organ transplant.

The utility of beta-D-glucan in the diagnosis of invasive aspergillosis remains unclear since it is a cell wall component of many fungi and it is not specific for *Aspergillus* species. Blood levels are often elevated in the setting of gut fungal translocation due to compromised intestinal epithelium due to diarrhea, GI Tract immune dysfunction or altered gut microbiota [69,70,71]. False-positive results can occur in those receiving albumin, and for hemodialysis with cellulose membranes, they can occur in those receiving IVIG and IV amoxicillin–clavulanic acid [72].

## 6. Imaging

The radiographic features of invasive pulmonary aspergillosis are variable and frequently nonspecific. The typical radiologic manifestations of IFD include nodules, masses, segmental or subsegmental consolidations, atelectasis, ground-glass opacities, a tree-in-bud pattern, cavities or pleural effusions [73]. Invasive pulmonary aspergillosis may appear as solitary or multiple pulmonary nodules/masses on chest X-rays or possibly wedge-like areas of ill-defined opacities. Chest X-rays have poor sensitivity and lack the ability to differentiate IPA from other etiologies of pneumonia, so a CT chest scan is the imaging modality of choice [74]. The classical findings in CT chest scans include multiple nodules with a “halo” sign, which is an area of ground-glass opacity surrounding a pulmonary nodule [74]. The ground-glass component represents angioinvasion and hemorrhage into the area surrounding the fungal nodule. An area of cavitation may develop in the nodule, which is classically described as an “air crescent sign” (Figure 2a,b).

Although both findings have been described as classical in invasive pulmonary aspergillosis, the most common presentation among immunocompromised patients is one or more macronodules in 94% of patients, while the halo sign is only present in approximately 61% of patients, with consolidations present in 30%, cavitary lesions in 20% of patients and, less commonly, with an air crescent sign (10%) [75]. The air crescent sign is a rare finding in early pulmonary aspergillosis, and it may not assist in the early diagnosis of IAPA. In neutropenic patients, it can be detected after the partial recovery of the neutrophil function [76]. These two classic findings described in immunocompromised patients are far less frequently encountered in non-neutropenic patients. Among 116 patients with non-neutropenic IPA, the most common findings included consolidation (47.4%) and cavities (47.4%), whereas the air crescent sign was detected in only 14.7% and the halo sign in 3.4% [37].

Despite being the imaging modality of choice, aiding in the early diagnosis of IPA, the CT findings are not 100% sensitive nor specific [74,77]. Some CT signs have demonstrated excellent sensitivity and specificity in select populations. The presence of a macronodule and/or consolidation or mass in the study by Park et al. had a sensitivity of 98% for IPA, while the presence of the hypodensity sign had excellent specificity for IA in the study by Horger et al., approaching a 100% offset by a low specificity of 30% [78,79]. However, most of these studies focus on organ transplant recipients or otherwise immunocompromised patients. The “halo” sign is important among neutropenic patients, but nonspecific for the diagnosis of IPA in other groups of patients [21]. No CT findings are specific enough to reliably differentiate IA in ICU patients where a significant overlap between the radiological features of the co-existing conditions, such as ARDS, COPD, atelectasis, etc., is present [7,12,21]. The overall sensitivity and specificity of CT imaging is comparable to that of thoracic MRI; however, it is far more widely available [74].

PET/CT can add value to anatomy-based studies, for example, CT and CXR. It can assist in detecting lesions outside the limited area that are included in traditional anatomy-based imaging, which is important in case of disseminated infections. Furthermore, serial images can aid in monitoring the response to treatment and aid in the decision to continue or stop antifungals [80,81,82,83,84,85].

## 7. Treatment of Invasive Aspergillosis

Currently, the triazoles, voriconazoles and isavuconazole are considered first-line treatment options for invasive aspergillosis in patients with hematological malignancies as per the ESCMID-ECMM-ERS guideline [86]. The IDSA guidelines, published in 2016, name voriconazole as the primary therapy of choice for most of the invasive aspergillus infections, whereas isavuconazole and liposomal amphotericin B are considered alternative therapies (Table 5).

Isavuconazole was shown to be non-inferior to voriconazole in the study by Maertens et al. and was noted to have fewer adverse events [88]. Alternative approaches include liposomal amphotericin B and itraconazole. Although combinations of drugs from different classes have been utilized frequently, randomized studies have not been conducted [89]. The combination of voriconazole plus low-dose anidulafungin revealed in vitro synergy against azole-resistant *A. fumigatus* [90]. The superiority of combination therapy over monotherapy has not been demonstrated but can be considered as salvage therapy in patients not responding to monotherapy [89,91,92,93]. Among echinocandins, micafungin and caspofungin are listed as salvage treatments of IA; however, caspofungin performance was suboptimal [94,95].

Voriconazole is considered the first-line option for patients without hematological malignancies as its use is associated with lower mortality [96,97]. Resistance to antifungals is an increasing global problem [98]. *Aspergillus* species can be intrinsically resistant to antifungals and can also develop secondary resistance during treatment [99,100]. According to CDC, about 7% of resistant isolates have been identified in the US, predominantly from patients with hematological malignancies and SOT recipients. The resistance to azoles among isolates of *A*. *fumigatus* in Europe continues to increase and poses a significant management challenge [99,101]. Resistance to azoles translates into higher mortality [102].

Antifungal susceptibility testing is recommended on clinically relevant Aspergillus isolates, especially when there is a lack of response to treatment and in patient groups or areas with known high levels of azole resistance [86]. IDSA recommends against routine antifungal susceptibility testing, unless there is a lack of response to treatment or for epidemiological purposes [87]. Some species are intrinsically resistant to certain antifungals (i.e., *A. terreus* is intrinsically resistant to amphotericin B, etc.), so species identification is necessary to guide treatment decisions [100]. MIC testing to determine susceptibility to azoles is carried out for resistance surveillance and clinical management. Local resistance rates guide the antifungal agent selection; if the local azole resistance rate exceeds 10%, the ESCMID-ECMM-ERS guideline recommends adding echinocandin or using liposomal amphotericin B [86]. A similar approach is recommended for isolates with high MICs to voriconazole [86].

Therapeutic drug monitoring is recommended for itraconazole, voriconazole, posaconazole and flucytosine to ensure both the safety and efficacy of the therapy [86]. There is limited data to routinely support isavuconazole therapeutic drug monitoring; however, it can be considered when there are concerns regarding treatment failure or drug interactions [103,104]. A study by Mikulska et al. suggested that significantly lower serum levels of isavuconazole were observed among ICU vs. non-ICU patients, which may necessitate therapeutic drug monitoring of isavuconazole in the ICU patients to ensure efficacy [103].

Novel antifungals in the pipeline have differing mechanisms of action and are being actively studied as treatment options for invasive aspergillosis [105,106]. Fosmanogepix is a first-in-class antifungal drug that inhibits the fungal glycosylphosphatidylinositol (GPI)-anchored wall-transfer protein 1 (Gwt1 protein), which affects the maturation and localization of the fungal wall mannoproteins, impacting the cell wall integrity and disrupting the processes of hyphal formation and biofilm formation. GPI has shown promise in azole-resistant aspergillosis and has a favorable side-effect profile [107,108,109,110,111]. The combination of fosmanogepix and Liposomal amphotericin B has shown promising results in in vitro models [112].

Opelconazole is a long-acting inhaled triazole, which achieves high concentrations in the respiratory tract and has limited systemic absorption which reduces adverse events and drug–drug interactions [113]. Olorofim, a reversible inhibitor dihydroorotate dehydrogenase, is active against azole- and amphotericin B-resistant *Aspergillus* species, including some of the non-*fumigatus Aspergillus* species [95,114,115]. A phase 3 study comparing Olorofim versus liposomal amphotericin B, followed by standard of care therapy (OASIS) in the management of invasive aspergillosis of the lower respiratory tract disease, is in progress (NCT05101187).

Ibrexafungerp, another newer antifungal, works by inhibiting (1,3)-β-D-glucan synthase in *Aspergillus*, *Candida* and other fungi [116]. It has demonstrated potent activity in vitro against *Aspergillus* species and demonstrated synergistic activity with voriconazole, isavuconazole and amphotericin B [117,118]. Finally, rezafungin, is a long-acting echinocandin, approved for candidemia and invasive candidiasis. It has demonstrated in vitro activity against *Aspergillus* spp, including azole-resistant species. However, it has not been studied in invasive aspergillosis and further studies are necessary [119,120,121].

## 8. Prophylaxis

Antifungal chemoprophylaxis is a standard of care in patients with prolonged neutropenia, acute myeloid leukemia or MDS, history of invasive aspergillosis per engraftment, GVHD after a follow-up SCT, rare cases of inherited immunodeficiency and select SOT. Prophylactic posaconazole reduces the incidence of invasive aspergillosis among patients with prolonged neutropenia and stem cell transplant with GVHD or AML. The value of prophylaxis to reduce the incidence of IA in non-neutropenic patients has not been established.

Currently, antifungals are not indicated for the prevention of IA in the ICU setting. Studies indicated that antifungal prophylaxis might be beneficial if the baseline incidence of IA is greater than 15–30%.

The Isavu-CAPA trial, investigating the use of isavuconazole for the prevention of CAPA was terminated early due to participant enrollment challenges [122]. According to the observational study by Hatzl et al., antifungal prophylaxis among ICU patients with COVID-19 was associated with significantly reduced CAPA incidence; however, it did not result in improved survival [123].

In 2021, Vanderbeke et al. published the results of a randomized, open-label, proof-of-concept trial evaluating posaconazole for the prevention of invasive pulmonary aspergillosis and critically ill influenza patients (POSA-FLU). Seven days of posaconazole prophylaxis were compared to no prophylaxis (standard of care, SOC) in these subjects. The incidence of IPA was not significantly reduced in the treatment arm when compared to the standard of care. However, this study was underpowered because most of the patients with IPA were excluded from the study as IPA was diagnosed within 48 h of ICU admission [124].

The retrospective study evaluating nebulized conventional amphotericin B as a means of prophylaxis of CAPA also showed a decreased incidence of CAPA in the prophylaxis groups; however, the mortality benefit was not demonstrated [125,126].

## 9. Conclusions

The diagnosis and management of IA in the ICU setting, despite it being more widely recognized by clinical providers, remains challenging. It contributes to an increase in mortality and poorer outcomes among patients in the ICU. It remains an under-recognized infection and awareness among providers remains low, contributing to the poor overall outcomes. Furthermore, variations in the definitions of IA in non-neutropenic patients further contribute to the insufficiently accurate estimation of the true incidence and prevalence of IA in the ICU.

It is essential to raise awareness among critical care physicians regarding the incidence of IA among ICU patients. In particular, the absence of “classical risk factors” may aid in the early diagnosis and appropriate treatment of these infections, which should lead to improved outcomes. Further studies are necessary in the ICU to aid in developing appropriate clinical guidelines and protocols to help navigate the approach to diagnosis and treatment of IA among these patients.

## Figures and Tables

**Figure 1 jof-11-00070-f001:**
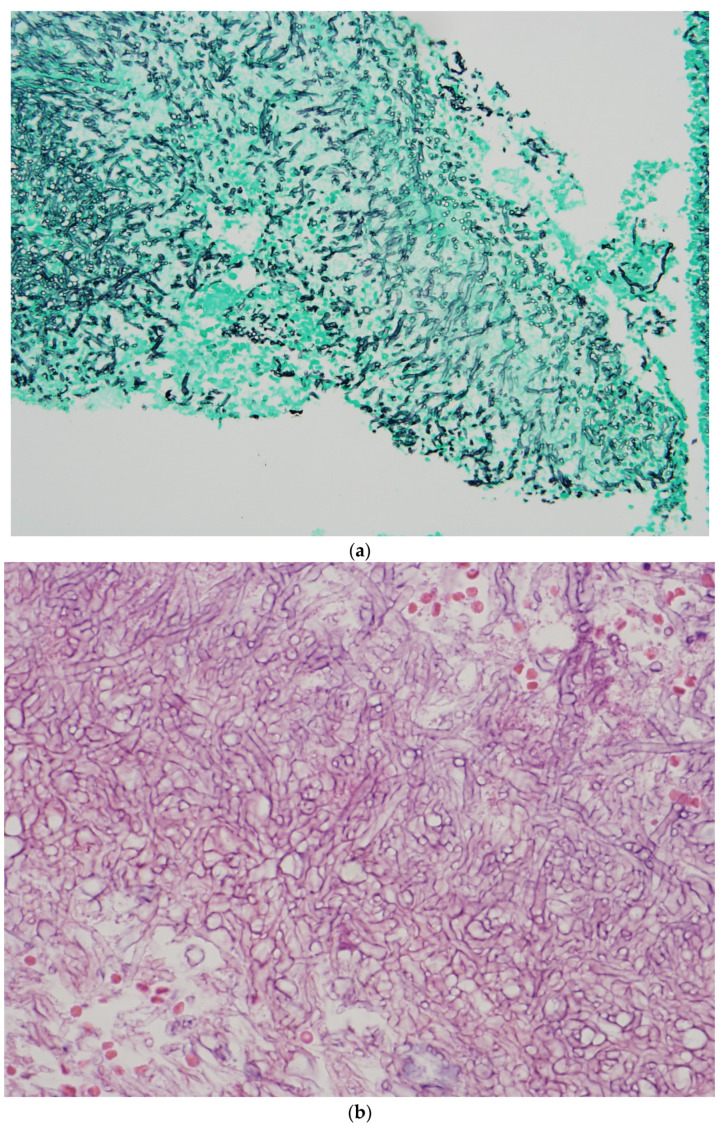
(**a**) Aspergillus in tissue, Grocott Methenamine stain (×250). (**b**) Aspergillus in tissue, H&E stain (×400).

**Figure 2 jof-11-00070-f002:**
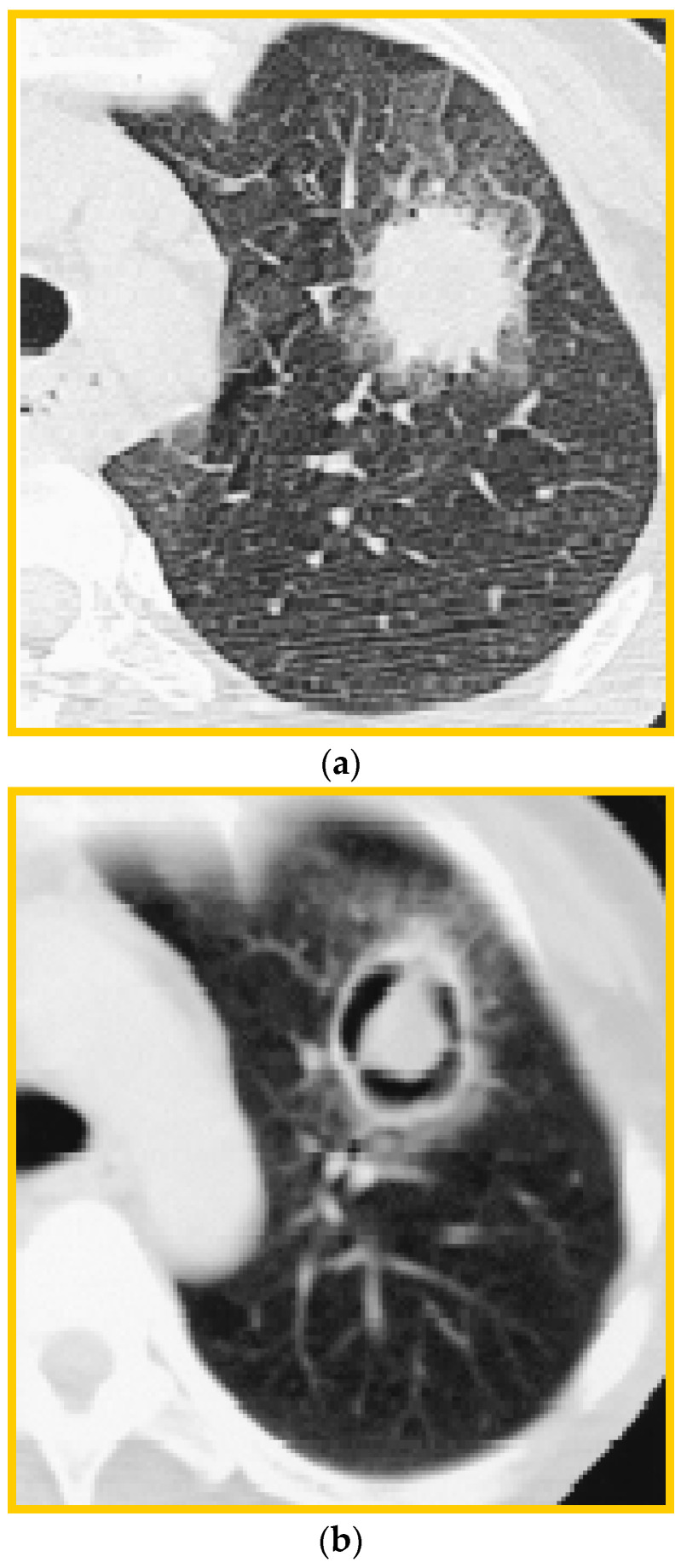
(**a**) Invasive pulmonary aspergillosis on HRCT. “Halo” sign. (**b**) Invasive pulmonary aspergillosis on HRCT. “Air crescent sign”. (HRCT = High-resolution Cat Scan).

**Table 1 jof-11-00070-t001:** Risk factors for invasive aspergillosis in the ICU [7].

High Risk	Intermediate Risk	Low Risk
Neutropenia (neutrophil count <500/mm^3^)	Prolonged treatment with corticosteroids before admission to the ICU	Severe burns
Hematologic malignancy	Autologous bone marrow transplantation	Other solid organ transplant recipients (e.g., heart, kidney or liver)
Allogeneic bone marrow transplantation	COPD	Steroid treatment with a duration of ≤7 days
	Liver cirrhosis with a duration of stay in the ICU >7 days	Prolonged stay in the ICU (>21 days)
	Solid organ cancer	Malnutrition
	HIV infection	Post-cardiac surgery status
	Lung transplantation	
	Systemic diseases requiring immunosuppressive therapy	

**Table 2 jof-11-00070-t002:** Clinical and radiological features of probable invasive aspergillosis in the ICU setting [7,12,21,22].

Pulmonary aspergillosis
The presence of one of the following patterns on a CT:
Dense, well-circumscribed lesions with or without a halo sign (typical in neutropenic patients). The halo sign is often not applicable in the ICU as the sign arrives too early (5 days before the onset of disease); not specific for *Aspergillus*
Air crescent sign—often not applicable in the ICU because it is obscured by atelectasis, ARDS and/or pleural effusion
Cavity
Wedge-shaped and segmental or lobar consolidation
Bronchopneumonia
Nodular disease
Tracheobronchitis
Tracheobronchial ulceration, pseudomembranes, nodule, plaque or eschar detected by bronchoscopy
Sino-nasal diseases
Acute localized pain (including pain radiating to the eye)
Nasal ulcer with black eschar
Extension from the paranasal sinus across bony barriers, including into the orbit
Central nervous system infection
One of the following two signs:
Focal lesions on imaging
Meningeal enhancement on MRI or CT

**Table 3 jof-11-00070-t003:** Mycological evidence supporting the diagnosis of probable invasive aspergillosis (based on updated EORTC/MSG criteria [21]).

Invasive pulmonary Aspergillosis
*Aspergillus* recovered by culture from sputum, BAL, bronchial brush or aspirate
Microscopical detection of fungal elements in sputum, BAL, bronchial brush or aspirate indicating *Aspergillus*
Tracheobronchitis
*Aspergillus* recovered by the culture of BAL or bronchial brush
Microscopic detection of fungal elements in BAL or bronchial brush indicating *Aspergillus*
Sino-nasal diseases
*Aspergillus* recovered by the culture of sinus aspirate samples
Microscopic detection of fungal elements in sinus aspirate samples indicating *Aspergillus*
Galactomannan antigen
Antigen detection plasma, serum, BAL or CSF
Any one of the following:
Single serum or plasma: ≥1.0
BAL fluid: ≥1.0
Single serum or plasma: ≥0.7 and BAL fluid ≥0.8
CSF: ≥1.0
*Aspergillus* PCR
Any one of the following:
○ Plasma, serum or whole blood from two or more consecutive PCR tests is positive
○ BAL fluid from two or more duplicate PCR tests positive
○ At least one positive PCR test in plasma, serum or whole blood and one positive PCR test in BAL fluid

**Table 5 jof-11-00070-t005:** Summary of recommendations for the treatment of Aspergillosis (from the IDSA-2016 guidelines [87]).

Therapy			
Condition	Primary	Alternative	Comments
IPA	Voriconazole (6 mg/kg IV every 12 h for 1 d, followed by 4 mg/kg IV every 12 h; oral therapy can be used at 200–300 mg every 12 h or weight-based dosing on a mg/kg basis); see text for pediatric dosing	Primary: Liposomal AmB (3–5 mg/kg/day IV), isavuconazole 200 mg every 8 h for 6 doses, then 200 mg daily. Salvage: ABLC (5 mg/kg/day IV), caspofungin (70 mg/day IV × 1, then 50 mg/day IV thereafter), micafungin (100–150 mg/day IV), posaconazole (oral suspension: 200 mg TID; tablet: 300 mg BID on day 1, then 300 mg daily; IV: 300 mg BID on day 1, then 300 mg daily), itraconazole suspension (200 mg PO every 12 h)	Primary combination therapy is not routinely recommended; the addition of another agent or the switch to another drug class for salvage therapy may be considered in individual patients; dosage in pediatric patients for voriconazole and for caspofungin is different than that of adults; limited clinical experience is reported with anidulafungin; the dosage of posaconazole in pediatric patients has not been defined
Invasive sinus aspergillosis	Similar to IPA	Similar to IPA	Surgical debridement as an adjunct to medical therapy
Tracheobronchial aspergillosis	Similar to IPA	Adjunctive inhaled AmB may be useful	Similar to IPA
Aspergillosis of the CNS	Similar to IPA	Similar to IPA Surgical resection may be beneficial in selected cases	This infection is associated with the highest mortality among all of the different patterns of IA; drug interactions with anticonvulsant therapy
Aspergillus infections of the heart (endocarditis, pericarditis and myocarditis)	Similar to IPA	Similar to IPA	Endocardial lesions caused by the Aspergillus species require surgical resection. Aspergillus pericarditis usually requires pericardiectomy
Aspergillus osteomyelitis and septic arthritis	Similar to IPA	Similar to IPA	Surgical resection of devitalized bone and cartilage is important for curative intent
Aspergillus peritonitis	Similar to IPA	Similar to IPA	Removal of peritoneal dialysis catheter is essential
Empiric and preemptive antifungal therapy	For empiric antifungal therapy, Liposomal AmB (3 mg/kg/day IV), caspofungin (70 mg day 1 IV and 50 mg/day IV thereafter), micafungin (100 mg day), voriconazole (6 mg/kg IV every 12 h for 1 day, followed by 4 mg/kg IV every 12 h; oral therapy can be used at 200–300 mg every 12 h or 3–4 mg/kg q 12 h)		Pre-emptive therapy is a logical extension of empiric antifungal therapy in defining a high-risk population with evidence of invasive fungal infection (e.g., pulmonary infiltrate or positive GM assay result)

Abbreviations: AML—acute myelogenous leukemia; ARF—acute respiratory failure; BALF—bronchoalveolar lavage fluid; CAPA—COVID-19-associated pulmonary aspergillosis; HM—hematological malignancy; HSCT—hematopoietic stem cell transplant; IFD—invasive fungal disease; mcfDNA—microbial cell-free DNA; mNGS—metagenomic next-generation sequencing; RUO-BAL KT—research-use-only Karius test on bronchoalveolar lavage fluid; SOT—solid organ transplant.

## Data Availability

No new data were created or analyzed in this study.

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
