# Peer review of "Invasive Aspergillosis in the Intensive Care Unit"

_jof, 2025, doi:10.3390/jof11010070_

Round 1

Reviewer 1 Report

Dear authors;

I would like to praise the manuscript presented to the JoF and the theme of aspergillosis in a hospital environment.

The manuscript presents quality information for a review.

I suggest that the tables be formatted according to the standards (with only the upper and lower lines). I would like to include figures in the manuscript (macro and micro morphology of fungi) in the introduction.

In item 6 (image): add some X-ray that helps us in the diagnosis of systemic mycosis.

The manuscript is a review, and for this reason, it could have illustrative figures for the article that will be published.

Kind regards;

Dear authors;

Please see my comments cited above.

Kind regards
—-

Author Response

I would like to praise the manuscript presented to the JoF and the theme of aspergillosis in a hospital environment.

The manuscript presents quality information for a review.

I suggest that the tables be formatted according to the standards (with only the upper and lower lines). I would like to include figures in the manuscript (macro and micro morphology of fungi) in the introduction.

In item 6 (image): add some X-ray that helps us in the diagnosis of systemic mycosis.

The manuscript is a review, and for this reason, it could have illustrative figures for the article that will be published.

Responses: 

  1. Unsure how to modify the tables??
  2. As you can see in the manuscript, we have add several figures and CT scans. All are the authors personal files.

Reviewer 2 Report

I found the article very interesting. The article give as a very complete revision of IA in ICU with the particularities in clinical presentation, diagnosis and its limitations very well described. 

It includes definition criteria, revision series, imagiology and treatment

I don't see any problem in english 

It seems to me a very complete revision of IA in ICU

I do not have any particular comment

Table 4 although includes all recent studies maybe is too long 

Author Response

Reviewer 2:

  • Lines 20–26: Please provide references or citations.

Response: corrected.

 Table 4 was left as is. We felt it would detract from the information being provided. However, if the editors think it is too extensive, we can try to delete it. 

  • Lines 34–56: In the Epidemiology section, be sure to mention the occurrence of different mould species - specifically, which Aspergillus species are most common in ICU patients, those with IAPA and CAPA, as well as patients with other underlying conditions. Are there any differences in species distribution among these groups?

Response: Addressed. Please see lines 47-52

  • Lines 43–44: Can you find more recent data?
  • Response: Lines 42-43 contain references to the articles from 2021 and 2023, and several from earlier years, such as 2004 and 2006. It provides both new and old data.

  • Lines 87–95: Briefly outline the key points of the Asp-ICU algorithm in a few bullet points.

Response: corrected. Please see lines 104-118.

  • Line 100: Mention Table 2 in the text.

Response: corrected.

  • Line 108: Mention Table 3 in the text.

Response: corrected.

  • Lines 109–113: Summarize the key points of the revised IA definitions in a few bullet points.

Response: addressed, please see lines 138-156.

  • Lines 124–146: Provide a brief description of the development and pathogenesis of IAPA, CAPA, and IA in general in the ICU.

Response: Corrected. Please see lines 167-181.

  • Lines 187–193: Provide references or citations.
  • Response: corrected. Please see lines 235-241.

  • Line 201: Write the genus name in italics.

Response: corrected.

  • Lines 210–213: Provide references or citations.
  • Response: corrected. Please see lines 266-269.

  • Lines 250–251: Provide references or citations.

Response: corrected.Please see lines 305-309.

  • Line 265: Mention Table 4 in the text.

Response: corrected.

  • Lines 285–312: Discuss the sensitivity and specificity of chest CT imaging and the importance of its timely use for the early detection of IA.
  • Response: corrected, see lines 377-389..

  • Line 318: Replace "Ambisome" with "liposomal amphotericin B."

Response: corrected.

  • Line 320: Explain the abbreviation "mcfDNA."

Response: corrected.

  • Lines 314–371: Highlight the importance and role of in vitro antifungal susceptibility testing and therapeutic monitoring of triazole concentrations in patients’ blood.
  • Response: corrected: please see lines 429-448.

Reviewer 2:

  • Lines 20–26: Please provide references or citations.

Response: corrected.

  • Lines 34–56: In the Epidemiology section, be sure to mention the occurrence of different mould species - specifically, which Aspergillus species are most common in ICU patients, those with IAPA and CAPA, as well as patients with other underlying conditions. Are there any differences in species distribution among these groups?

Response: Addressed. Please see lines 47-52

  • Lines 43–44: Can you find more recent data?
  • Response: Lines 42-43 contain references to the articles from 2021 and 2023, and several from earlier years, such as 2004 and 2006. It provides both new and old data.

  • Lines 87–95: Briefly outline the key points of the Asp-ICU algorithm in a few bullet points.

Response: corrected. Please see lines 104-118.

  • Line 100: Mention Table 2 in the text.

Response: corrected.

  • Line 108: Mention Table 3 in the text.

Response: corrected.

  • Lines 109–113: Summarize the key points of the revised IA definitions in a few bullet points.

Response: addressed, please see lines 138-156.

  • Lines 124–146: Provide a brief description of the development and pathogenesis of IAPA, CAPA, and IA in general in the ICU.

Response: Corrected. Please see lines 167-181.

  • Lines 187–193: Provide references or citations.
  • Response: corrected. Please see lines 235-241.

  • Line 201: Write the genus name in italics.

Response: corrected.

  • Lines 210–213: Provide references or citations.
  • Response: corrected. Please see lines 266-269.

  • Lines 250–251: Provide references or citations.

Response: corrected.Please see lines 305-309.

  • Line 265: Mention Table 4 in the text.

Response: corrected.

  • Lines 285–312: Discuss the sensitivity and specificity of chest CT imaging and the importance of its timely use for the early detection of IA.
  • Response: corrected, see lines 377-389..

  • Line 318: Replace "Ambisome" with "liposomal amphotericin B."

Response: corrected.

  • Line 320: Explain the abbreviation "mcfDNA."

Response: corrected.

  • Lines 314–371: Highlight the importance and role of in vitro antifungal susceptibility testing and therapeutic monitoring of triazole concentrations in patients’ blood.
  • Response: corrected: please see lines 429-448.

Reviewer 3 Report

The authors have written a very nice, concise review article on invasive aspergillosis in the ICU. I commend the authors for mentioning all the tables in the text and not just some of them. The information is well organised and presented in the tables. However, I would suggest including only the most important points from the tables in the text, in the form of bullet points. I also miss a description of the importance and role of in vitro antifungal susceptibility testing and therapeutic monitoring of triazole concentrations in patients’ blood.

  • Lines 20–26: Please provide references or citations.
  • Lines 34–56: In the Epidemiology section, be sure to mention the occurrence of different mould species - specifically, which Aspergillus species are most common in ICU patients, those with IAPA and CAPA, as well as patients with other underlying conditions. Are there any differences in species distribution among these groups?
  • Lines 43–44: Can you find more recent data?
  • Lines 87–95: Briefly outline the key points of the Asp-ICU algorithm in a few bullet points.
  • Line 100: Mention Table 2 in the text.
  • Line 108: Mention Table 3 in the text.
  • Lines 109–113: Summarise the key points of the revised IA definitions in a few bullet points.
  • Lines 124–146: Provide a brief description of the development and pathogenesis of IAPA, CAPA, and IA in general in the ICU.
  • Lines 187–193: Provide references or citations.
  • Line 201: Write the genus name in italics.
  • Lines 210–213: Provide references or citations.
  • Lines 250–251: Provide references or citations.
  • Line 265: Mention Table 4 in the text.
  • Lines 285–312: Discuss the sensitivity and specificity of chest CT imaging and the importance of its timely use for the early detection of IA.
  • Line 318: Replace "Ambisome" with "liposomal amphotericin B."
  • Line 320: Explain the abbreviation "mcfDNA."
  • Lines 314–371: Highlight the importance and role of in vitro antifungal susceptibility testing and therapeutic monitoring of triazole concentrations in patients’ blood.

Author Response

Reviewer 2:

  • Lines 20–26: Please provide references or citations.

Response: corrected.

  • Lines 34–56: In the Epidemiology section, be sure to mention the occurrence of different mould species - specifically, which Aspergillus species are most common in ICU patients, those with IAPA and CAPA, as well as patients with other underlying conditions. Are there any differences in species distribution among these groups?

Response: Addressed. Please see lines 47-52

  • Lines 43–44: Can you find more recent data?
  • Response: Lines 42-43 contain references to the articles from 2021 and 2023, and several from earlier years, such as 2004 and 2006. It provides both new and old data.

  • Lines 87–95: Briefly outline the key points of the Asp-ICU algorithm in a few bullet points.

Response: corrected. Please see lines 104-118.

  • Line 100: Mention Table 2 in the text.

Response: corrected.

  • Line 108: Mention Table 3 in the text.

Response: corrected.

  • Lines 109–113: Summarize the key points of the revised IA definitions in a few bullet points.

Response: addressed, please see lines 138-156.

  • Lines 124–146: Provide a brief description of the development and pathogenesis of IAPA, CAPA, and IA in general in the ICU.

Response: Corrected. Please see lines 167-181.

  • Lines 187–193: Provide references or citations.
  • Response: corrected. Please see lines 235-241.

  • Line 201: Write the genus name in italics.

Response: corrected.

  • Lines 210–213: Provide references or citations.
  • Response: corrected. Please see lines 266-269.

  • Lines 250–251: Provide references or citations.

Response: corrected.Please see lines 305-309.

  • Line 265: Mention Table 4 in the text.

Response: corrected.

  • Lines 285–312: Discuss the sensitivity and specificity of chest CT imaging and the importance of its timely use for the early detection of IA.
  • Response: corrected, see lines 377-389..

  • Line 318: Replace "Ambisome" with "liposomal amphotericin B."

Response: corrected.

  • Line 320: Explain the abbreviation "mcfDNA."

Response: corrected.

  • Lines 314–371: Highlight the importance and role of in vitro antifungal susceptibility testing and therapeutic monitoring of triazole concentrations in patients’ blood.
  • Response: corrected: please see lines 429-448.